# Chlorophyll inversion in rice based on visible light images of different planting methods

He Jing[1]*, Wang Bin[2], He Jiachen[3]

1 School of Geography and Planning, Chengdu University of Technology, Chengdu, China, 2 Housing and Urban-Rural Development Bureau, Leshan, Sichuan, China, 3 National Chengdu Agricultural Science and Technology Center, Chengdu, China

* xiao00yao@163.com

## Abstract

As a key substance for crop photosynthesis, chlorophyll content is closely related to crop growth and health. Inversion of chlorophyll content using unmanned aerial vehicle (UAV) visible light images can provide a theoretical basis for crop growth monitoring and health diagnosis. We used rice at the tasseling stage as the research object and obtained UAV visible orthophotos of two experimental fields planted manually (experimental area A) and mechanically (experimental area B), respectively. We constructed 14 vegetation indices and 15 texture features and utilized the correlation coefficient method to analyze them comprehensively. Then, four vegetation indices and four texture features were selected from them as feature variables to be added into three models, namely, K-neighborhood (KNN), decision tree (DT), and AdaBoost, respectively, for inverting chlorophyll content in experimental areas A and B. In the KNN model, the inversion model built with BGRI as the independent variable in region A has the highest accuracy, with $R^2$ of 0.666 and RSME of 0.79; the inversion model built with RGRI as the independent variable in region B has the highest accuracy, with $R^2$ of 0.729 and RSME of 0.626. In the DT model, the inversion model built with B-variance as the independent variable in region A has the highest accuracy, with $R^2$ of 0.840 and RSME of 0.464; the inversion model built with G-mean as the independent variable in region B has the highest accuracy, with $R^2$ of 0.845 and RSME of 0.530. In the AdaBoost model, the inversion model built with R-skewness as the independent variable in region A has the highest accuracy, with $R^2$ of 0.826 and RSME of 0.642; the inversion model established with g as the independent variable in area B had the highest accuracy, with $R^2$ of 0.879 and RSME of 0.599. In the comprehensive analysis, the best inversion models for experimental areas A and B were B-variance-decision tree and g-AdaBoost, respectively, whose models can quickly and accurately carry out the inversion of chlorophyll content of rice, and provide a theoretical basis for the monitoring of the crop's growth and health under different cultivation methods.

## 1. Introduction

Rice is one of the most widely grown food crops worldwide and is important for global food security and economic development. Accurate diagnosis of chlorophyll content is of great

**Data availability statement:** All relevant data are within the manuscript and its Supporting Information files.

**Funding:** Key Laboratory of the Evaluation and Monitoring of Southwest Land Resources (Ministry of Education) (TDSYS202406 to H.J), and the Chengdu City Technology Innovation R&D Project (2022-YF05-01090-SN to H.J)

**Competing interests:** The authors have declared that no competing interests exist.

significance for crop growth monitoring, nutritional diagnosis, and crop yield estimation, which is an important direction for the development of precision agriculture. By determining the chlorophyll content of rice during the spiking period, it is possible to determine whether rice growth is healthy or not, to intervene earlier in the growth of rice during the critical period, and to predict its future yield, thus providing a basis for subsequent crop management [1]–[3]. Traditional chlorophyll content diagnostic methods mainly include spectrophotometers, fluorescence analysis, and atomic absorption. Although these methods are accurate, they are cumbersome and have many steps, and they can damage crop growth and affect the subsequent growth and development of the crop [4,5]. In response to the shortcomings of traditional chlorophyll measurement methods, instruments have been developed for the rapid, non-destructive determination of the relative chlorophyll content of crops. For example, the SPAD chlorophyll handheld meter in Japan. In recent years, with the rapid development of remote sensing technology, unmanned aerial vehicle (UAV) remote sensing has been widely used in crop chlorophyll content inversion with its fast and flexible, high spatial resolution and good timeliness [6]. Cao et al. aimed at the problem of unclear characteristics of the red edge position in the unmanned aerial hyperspectral inversion of the chlorophyll content of Northeast Japonica rice and determined the position of the red edge of the hyperspectral image of the rice canopy by using six methods, such as linear extrapolation, linear four-point interpolation, etc., which finally showed that linear extrapolation-limit learner and linear extrapolation-logarithmic curve model inversion were more effective [7]. Wang et al. took jujube trees at the fruit-sitting stage as the research object, used UAV visible light images to monitor the SPAD value of jujube tree canopy at the field scale, and preferred vegetation indices to construct univariate regression, multivariate stepwise regression, and random forest regression models for estimating the SPAD value of jujube tree canopy, and concluded that it is feasible to estimate the SPAD value of jujube tree canopy by using UAV visible light remote sensing images [8]. Qiao et al. used a UAV carrying a multispectral sensor to collect visible and near-infrared images of maize canopy at the jointing stage under six fertilizer application levels to obtain the simple vegetation index, modified vegetation index, and functional vegetation index and established a model for detecting the chlorophyll content of maize canopies by using the partial least squares regression and random forest algorithms, and the accuracy of inversion models could reach 0.753 at the highest level [9].

Visible light imagery has the advantages of low cost, high spatial resolution, and simple data processing, which further reduces the need for wavelength-remote sensing imagery, thus greatly reducing the difficulty of image acquisition [10]. The spectral features of UAV visible remote sensing images reflect the visual characteristics of the images, and different features have unique spectral features, and the stability of the spectral features of the images is very good, but they have less spectral information and lack depth information [11]. With the continuous progress of UAV visible remote sensing imaging technology, the spatial and spectral resolution of feature images has been significantly improved, enabling more and more feature details to be clearly presented [12]. For this reason, after acquiring spectral features, we need to use texture features to describe more comprehensive and accurate feature information to improve the accuracy of target detection [13]. Qu et al. and Liu used UAV visible light images to extract the best variables from the constructed vegetation index to participate in the inversion model operation, which effectively realized the inversion of crop chlorophyll content [14,15]. However, they neglected the rich texture features of visible light images.

In recent years, the mechanization of rice production has shown a rapid development trend, the proportion of manual planting has decreased, and mechanized rice transplanting is gradually replacing traditional manual transplanting. As far as rice planting is concerned, replacing the traditional manual planting with the simple and efficient planting method

represented by mechanical transplanting is an important way to improve the efficiency of rice production and reduce the cost of human labor [16]. Numerous scholars have evaluated rice yield and economic efficiency [17], microbial populations [18], pest and weed pest occurrence characteristics [19,20], nitrogen utilization efficiency [21], and greenhouse gas emissions [22] under different planting methods, and concluded that different planting methods have different rice yield levels, nutrient uptake and utilization, and economic benefits have certain differences. However, relatively few studies have been conducted to evaluate the chlorophyll content of rice paddies under different planting methods in the plains.

Therefore, this paper proposes a chlorophyll inversion model that synergizes visible spectral information and texture features, which can invert the chlorophyll content of rice more accurately and efficiently. We extracted the spectral information and texture features of UAV visible images, constructed 14 spectral indices and 15 texture features, and selected the feature variables that were highly correlated with the measured rice chlorophyll content as the input variables of the model using correlation analysis. We compared the inversion results of these two parameters in three different models, namely, K-neighborhood (KNN), Decision Tree (DT), and AdaBoost, and evaluated the performance differences among these three inversion models in predicting the chlorophyll content of rice, and selected the best inversion model. In addition, to explore the applicability of the models in different planting types of farmland, we selected two representative planting types of farmland (manual planting and mechanical planting). We tested the inversion effects of the three models in these two types of farmland and deeply explored the applicable conditions of the models and the effects of different planting methods on the inversion effects of rice chlorophyll content in the plain area. This will provide a low-cost and high-precision monitoring method and reference basis for the development process of mechanized agriculture.

## 2. Materials and methods

### 2.1. Data acquisition

The test area was located in the rice production area of Yao Du Town, Qingbaijiang District, Chengdu City, Sichuan Province, and the data were collected on August 2, 2021 (Rice is in the tasseling stage, which is a critical period for screening high-quality rice grains, and rice begins to enter the reproductive growth stage after tasseling) from 11:30 to 14:20, and the weather was sunny and cloudless at the time of data collection. The data were collected using a DJI Elf 4 Pro-UAV, flying at an altitude of 100 m with 80% overlap in both heading and lateral directions; 10 positioning plates were deployed in the flight area as image control points; and finally, orthophotos were generated using Pix4D. The location map of the test area is shown in Fig 1. Two experimental areas, A and B, were selected as the actual measurement areas of rice chlorophyll content during the tasseling period of this experiment (see Fig 1), with experimental area A being the area of rice planted artificially and experimental area B being the area of mechanical planting. Chlorophyll measurement data were obtained using the Japanese SPAD-502 Plus handheld chlorophyll meter, and data were collected randomly at 60 measurement points (120 in total) in areas A and B, respectively. Each point was collected five times at different positions of the fully expanded leaves of rice, and the average value was taken as the chlorophyll value on the point. The measured data in each region were randomly assigned in the ratio of 7.5:2.5 as the modeling set and validation set to participate in the inverse model operation.

### 2.2. Feature parameter construction and selection

This experiment uses vegetation indices and texture features as characteristic parameters for the inversion of rice chlorophyll content from UAV visible light images.

The vegetation index is a kind of index reflecting the difference between the reflectance of green vegetation and soil, water, etc. in the visible and near-infrared bands, which is obtained by mathematically combining different bands, and it is a feature used to enhance the information of vegetation [23]. Based on previous studies, we constructed a total of 14 visible light vegetation indices, whose formulas and sources are detailed in Table 1.

Texture features are similar shapes containing strong or weak regularity in an image, it is a common visual phenomenon that repeats local structures or rules of arrangement on an image [30] and is an important feature of visible light images. Visible light images have three bands, red (R), green (G), and blue (B), and by performing second-order probabilistic statistical filtering on the grey-scale maps of these three bands, five texture features are obtained for each of the three bands: variance, skewness, mean, entropy, and range of values (datarange). Thus we extracted 15 texture features as detailed in Table 2.

In this paper, we used the correlation coefficient method to select the vegetation indices with textural features. The correlation coefficient is a measure of the linear correlation between two variables, and common coefficients used to describe the correlation between variables include Pearson, Spearman, Kendall, Polychoric, Tetrachoric, and Polyserial. Pearson correlation coefficient [31] can numerically measure the correlation between two

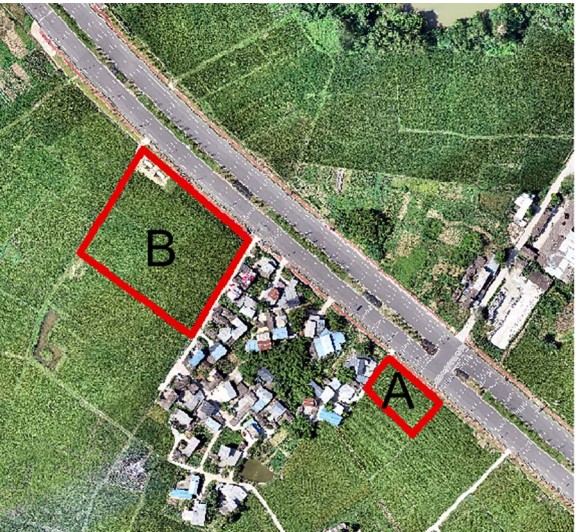

**Fig 1. Bitmap of test area.**

**Table 1. Vegetation index.**

| Vegetation index | Formulas | Source | Vegetation index | Formulas | Source |
|---|---|---|---|---|---|
| R | | * | G-R | G-R | * |
| G | | * | G-B | G-B | * |
| B | | * | GRRI | G/R | [24] |
| r | R/(R+G+B) | * | RGRI | R/G | [26] |
| g | G/(R+G+B) | [25] | BGRI | B/G | [27] |
| b | B/(R+G+B) | | RBRI | R/B | [28] |
| R-B | R-B | * | GBRI | G/B | [29] |

Note: 1. R, G and B are the pixel values of the corresponding bands of the visible light image

**Table 2. Texture features.**

| Wave band | Texture feature types |
|---|---|
| R | R-variance, R-skewness, R-mean, R-entropy, R-datarange |
| G | G-variance, G-skewness, G-mean, G-entropy, G-datarange |
| B | B-variance, B-skewness, B-mean, B-entropy, B-datarange |

continuous variables and its results are directional, so this paper chose to use Pearson correlation coefficient. The range of its value is between -1 and 1. The larger the absolute value, the better the correlation, when $P < 0.01$ indicates a highly significant correlation, and $P < 0.05$ indicates a significant correlation [32].

## 2.3. Inversion modelling

Machine learning models are highly adaptable and can handle a variety of complex and variable data. In the inversion of chlorophyll content in rice, machine learning algorithms can accurately estimate the chlorophyll level of rice and provide accurate guidance for agricultural production. Yun et al. used univariate regression, multivariate regression, and BP neural network algorithm based on principal component analysis to construct the maize SPAD estimation model, screen the optimal model and test it, respectively [33]. Liu et al. used deep neural networks and convolutional neural networks to establish the relationship model between color information of peanut leaf digital image and leaf chlorophyll content, which provides a theoretical basis for the rapid diagnosis of peanut fertilizer status in production. In this paper, we use three regression models, KNN, DT and AdaBoost [34].

KNN is a basic machine learning algorithm that performs classification or regression by measuring the distance between different data points. Its operation procedure is to calculate the distance between the sample to be tested and other known samples, then sort the calculated distances, select the k samples with the smallest distance from the sample to be tested, calculate the average value of the attributes of the k samples and assign it to the sample to be tested [35]. The KNN algorithm has the advantages of being easy to understand, easy to implement, and there is no specific restriction on the selection of features, and it can deal with a variety of types of data.

Decision tree is a typical method of data feature classification and regression analysis, the results of which are intuitive and easy to understand, the tree structure demonstrates the decision-making process without the need for complex pre-processing or conversion of data. It consists of multiple nodes and edges, and there are two types of nodes: internal nodes and leaf nodes, where the internal nodes represent a feature or attribute of a variable, and the leaf nodes represent a class, while the leaf nodes at the end of the DT represent the final decision results, and the other nodes correspond to the judgment rules of the attributes [36]. Decision trees have a simple structure, can be constructed with a small amount of data to build a model, and are suitable for regression prediction with small sample data sets.

AdaBoost is an integrated learning algorithm that builds on the framework of the Boosting algorithm by iteratively training multiple weak classifiers and adjusting the weights of the samples according to the performance of each classifier, to gradually construct a strong classifier. During the training process, the weight of misclassified samples will gradually increase, enabling the algorithm to pay more attention to these difficult-to-classify samples. The final strong classifier is a weighted combination of multiple weak classifiers, where the weight of

each weak classifier depends on its classification performance on the training set. The use of the AdaBoost algorithm can effectively avoid overfitting, and the algorithm has a strong learning ability and is not easy to over-adaptation phenomenon [37–39].

## 2.4. Evaluation indicators

We use the coefficient of determination ($R^2$) and the root mean square error (RMSE) to judge the reliability and accuracy of the model. Where a larger value of $R^2$ indicates a better fitting accuracy of the model inversion results, and a smaller value of RSME indicates a higher predictive accuracy of the model [40].

# 3. Results and analyses

## 3.1. Data analysis

The distribution of weeds around the paddy field can be clearly seen from the UAV orthophoto image in Fig 2 (the part drawn by the red line). However, it is difficult to visually identify the weeds in the paddy field through the images. Therefore, we can invert the chlorophyll content of the experimental area and combine it with the chlorophyll results of the weed area to analyze whether there are a large number of weeds in the paddy field and prepare for the subsequent weed control work.

Chlorophyll data analysis: Chlorophyll values measured for rice at the tasseling stage within the experimental areas A and B were statistically analyzed and are shown in Table 3. From Table 3, it can be seen that the measured chlorophyll values ranged from 37.62-43.32

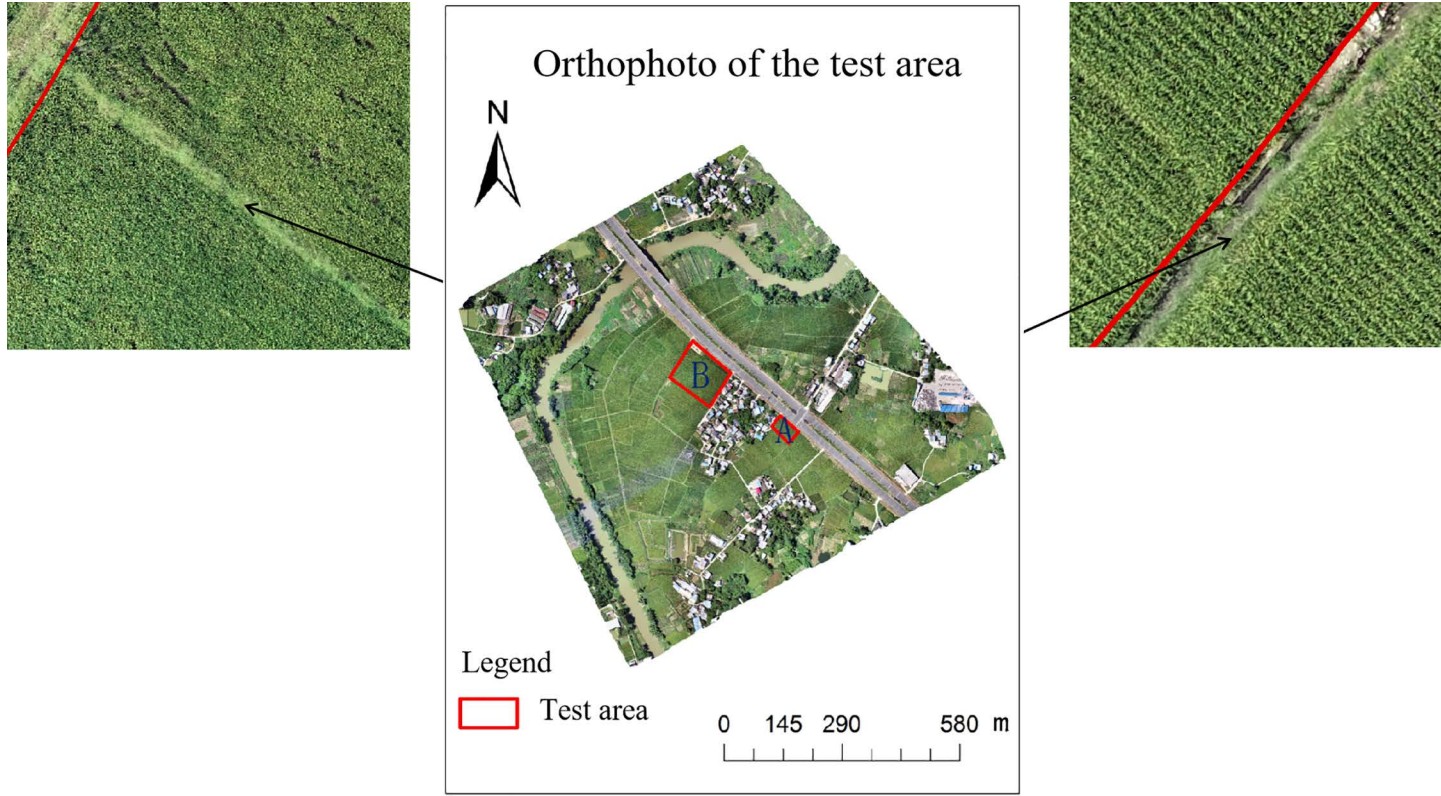

**Fig 2. Orthophoto of the test area.**

in Region A and 36.53-45.18 in Region B, indicating that the growth in Region A was more homogeneous than that in Region B. In terms of variance and standard deviation, rice chlorophyll values were less discrete in Region A than in Region B, with less variation in rice growth.

## 3.2. Feature selection

The correlation analysis of vegetation index, texture characteristics and chlorophyll measured values in areas A and B is shown in Fig 3, which shows the absolute value of the correlation coefficient. As can be seen from Fig 3a, the absolute value of the correlation coefficient between most vegetation indices and chlorophyll content in experimental area A is greater than 0.522. Combining the results of the original correlation analysis, four parameters, g, RGRI, BGRI and GBRI, were selected as characteristic variables, of which g and GBRI were positively and significantly correlated with correlation coefficients of 0.755 and 0.721, respectively; RGRI and BGRI were negatively and highly significantly correlated with correlation coefficients of -0.746 and -0.718, respectively. Most of the correlation coefficients between vegetation indices and chlorophyll content in experimental area B were greater than 0.42 in absolute value, and four parameters, g, G-R, RGRI and B, were selected as characteristic variables, of which RGRI was positively and significantly correlated, with a correlation coefficient of 0.58, while three, g, G-R and B, were negatively and significantly correlated, with correlation coefficients of -0.74, -0.6 and -0.55, respectively. Combining the correlation of vegetation indices in the two areas shows that more vegetation indices are negatively correlated with chlorophyll. From Fig. 3b, it can be seen that the absolute value of correlation coefficients of most textural features in the test area A is greater than 0.602. Four parameters, R-skewness, G-skewness, B-variance and B-skewness, were selected as feature variables, of which B-variance was positively and significantly correlated

**Table 3. Orthophoto of the test area.**

| Test area | Number of samples | Minimum value | Maximum value | Mean value | Variance | Standard deviation |
|---|---|---|---|---|---|---|
| A | 60 | 37.62 | 43.32 | 39.91 | 1.47 | 1.21 |
| B | 60 | 36.53 | 45.18 | 41.48 | 3.92 | 1.98 |

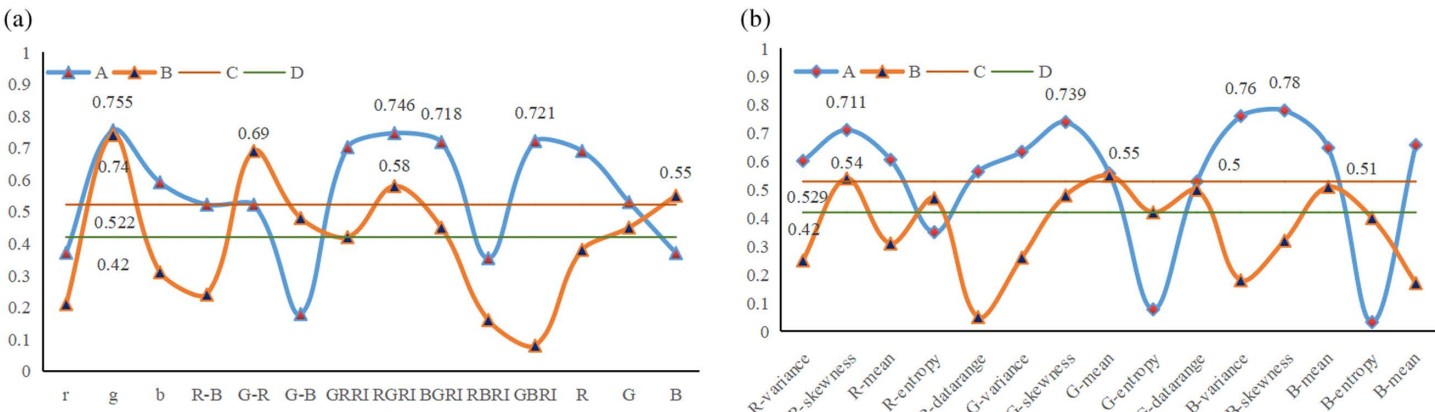

**Fig 3. Absolute value diagram of correlation coefficient.** Note: 1. A is the absolute coefficient of regional correlation of a, B is the absolute coefficient of regional correlation of B, C and D are the datum lines of a and B respectively, and their values are determined by extremely significant correlation; 2. * * indicates a very significant correlation.

with a correlation coefficient of 0.76; R-skewness, G-skewness and B-skewness were negatively and significantly correlated, with a correlation coefficient of 0.76; R- skewness, G-skewness and B-skewness were negatively and significantly correlated, with correlation coefficients of -0.711, -0.739 and -0.78, respectively. The absolute value of correlation coefficients of most texture features in experimental area B was greater than 0.42. Four parameters, R-skewness, G-mean, G-datarange and B-mean, were selected as feature variables, of which R-skewness and G-datarange were positively and significantly correlated, with correlation coefficients of 0.54 and 0.5, respectively; G-mean and B-mean were negatively and significantly correlated, with correlation coefficients of -0.55 and -0.51, respectively. The correlation of texture features in the two areas shows that more texture features are negatively correlated with chlorophyll. Combining the above analyses, the correlation between vegetation index, texture features and chlorophyll in test area A was better than that in test area B, and in general there were more negative than positive significant correlations.

### 3.3. Inversion modelling

Vegetation indices and texture characteristic factors selected from experimental areas A and B were input into KNN, DT and AdaBoost models as independent variables, respectively, to establish the chlorophyll inversion model, and the three models with the best results were selected from the inversion results to participate in the subsequent comparison.

**3.3.1. K-Nearest neighbors inversion model.** The results of the KNN inversion model are shown in Table 4, which shows that the inversion results of test area B are better than that of test area A. Moreover, the vegetation index, which is involved as an independent variable in the modeling of the two areas, obtains better inversion results than the texture features. The highest accuracy of both the modelling and validation sets of the inversion model for test area A was the inversion model built with BGRI as the independent variable, with $R^2$ of 0.583 and RSME of 0.720 for the modeling set, and $R^2$ of 0.666 and RSME of 0.79 for the validation set. The highest accuracy of both the modeling and validation sets of the B inversion model in the test area was for the inversion model built with RGRI as the independent variable, with $R^2$ of 0.774 for the modeling set and RSME of 0.760 for the validation set, and $R^2$ of 0.729 for the validation set and RSME of 0.626 for the validation set.

**3.3.2. Decision tree inversion model.** The results of the DT inversion model are shown in Table 5, which shows that the DT inversion model established with the vegetation index and texture features as independent variables achieved better inversion results in both A and B areas, and the inversion accuracies were similar in the two experimental areas of A and B. In both regions, the best inversion results were achieved by participating in the DT inversion model with texture features as independent variables, indicating that better results can be achieved by using texture features as independent variables for rice chlorophyll inversion with the DT as the inversion model. The highest accuracy in both modeling and validation sets of the inversion models in the test area A

**Table 4. KNN inversion model results.**

| Test area | Independent variable | Modeling set | | Validation set | |
|---|---|---|---|---|---|
| | | R² | RSME | R² | RSME |
| A | BGRI | 0.583 | 0.720 | 0.666 | 0.790 |
| | GBRI | 0.553 | 0.810 | 0.640 | 0.790 |
| | R-skewness | 0.425 | 0.845 | 0.651 | 0.826 |
| B | RGRI | 0.774 | 0.760 | 0.729 | 0.626 |
| | G-datarange | 0.575 | 1.393 | 0.624 | 0.926 |
| | g | 0.502 | 1.349 | 0.368 | 1.200 |

was the inversion model built with B-variance as the independent variable, with $R^2$ of 0.884 and RSME of 0.341 for the modeling set, and $R^2$ of 0.840 and RSME of 0.464 for the validation set. The highest accuracy of the modeling set of the B inversion model in the test area is the inversion model constructed with RGRI as the independent variable, with a modeling set $R^2$ of 0.846 and RSME of 0.254; the highest accuracy of the validation set is the inversion model built with G-mean as the independent variable, with a validation set $R^2$ of 0.845 and RSME of 0.53.

**3.3.3. AdaBoost inversion model.** The results of the AdaBoost inversion model are shown in Table 6, which shows that the inversion model built with the vegetation index and texture features as independent variables achieved better inversion results in both A and B regions. In both areas, the best inversion results were achieved by participating in the AdaBoost inversion model with texture features as independent variables. The results showed that the inversion of rice chlorophyll using AdaBoost as the inversion model with texture features as independent variables could achieve better results. And the inversion result of test area B was better than that of test area A. The highest accuracy of both the modeling and validation sets of the inversion model for test area A was the inversion model built with R-skewness as the independent variable, with $R^2$ of 0.801 and RSME of 0.497 for the modeling set, and $R^2$ of 0.826 and RSME of 0.642 for the validation set. The highest accuracy in the modeling set of the B inversion model in the test area is the inversion model constructed with G-mean as the independent variable, with a modeling set $R^2$ of 0.880 and RSME of 0.477; the highest accuracy in the validation set is the inversion model constructed with g as the independent variable, with a validation set $R^2$ of 0.879 and RSME of 0.599.

## 3.4. Comparative analysis of inversion models

Fig 4 shows the inversion results with the highest accuracy for the validation set of regression models in test areas A and B. As can be seen from the results of the inversion in test area A in Fig. 4, the best inversion result among the three model optimal inversion results is the B-variance-DT model constructed with the texture feature B-variance as the independent variable and the DT as the inversion model, and its prediction results and the measured results fit the equation of

**Table 5. Decision tree inversion model results.**

| Test area | Independent variable | Modeling set | | Validation set | |
|---|---|---|---|---|---|
| | | R² | RSME | R² | RSME |
| A | B-variance | 0.884 | 0.341 | 0.840 | 0.464 |
| | G-skewness | 0.698 | 0.631 | 0.790 | 0.582 |
| | RGRI | 0.801 | 0.497 | 0.780 | 0.655 |
| B | G-mean | 0.832 | 0.876 | 0.845 | 0.530 |
| | RGRI | 0.846 | 0.254 | 0.788 | 0.613 |
| | R-skewness | 0.668 | 1.231 | 0.775 | 0.715 |

**Table 6. AdaBoost inversion model results.**

| Test area | Independent variable | Modeling set | | Validation set | |
|---|---|---|---|---|---|
| | | R² | RSME | R² | RSME |
| A | R-skewness | 0.801 | 0.497 | 0.826 | 0.642 |
| | B-variance | 0.696 | 0.614 | 0.805 | 0.617 |
| | g | 0.778 | 0.525 | 0.790 | 0.565 |
| B | g | 0.802 | 0.663 | 0.879 | 0.599 |
| | G-datarange | 0.730 | 0.881 | 0.836 | 0.565 |
| | G-mean | 0.880 | 0.477 | 0.819 | 0.643 |

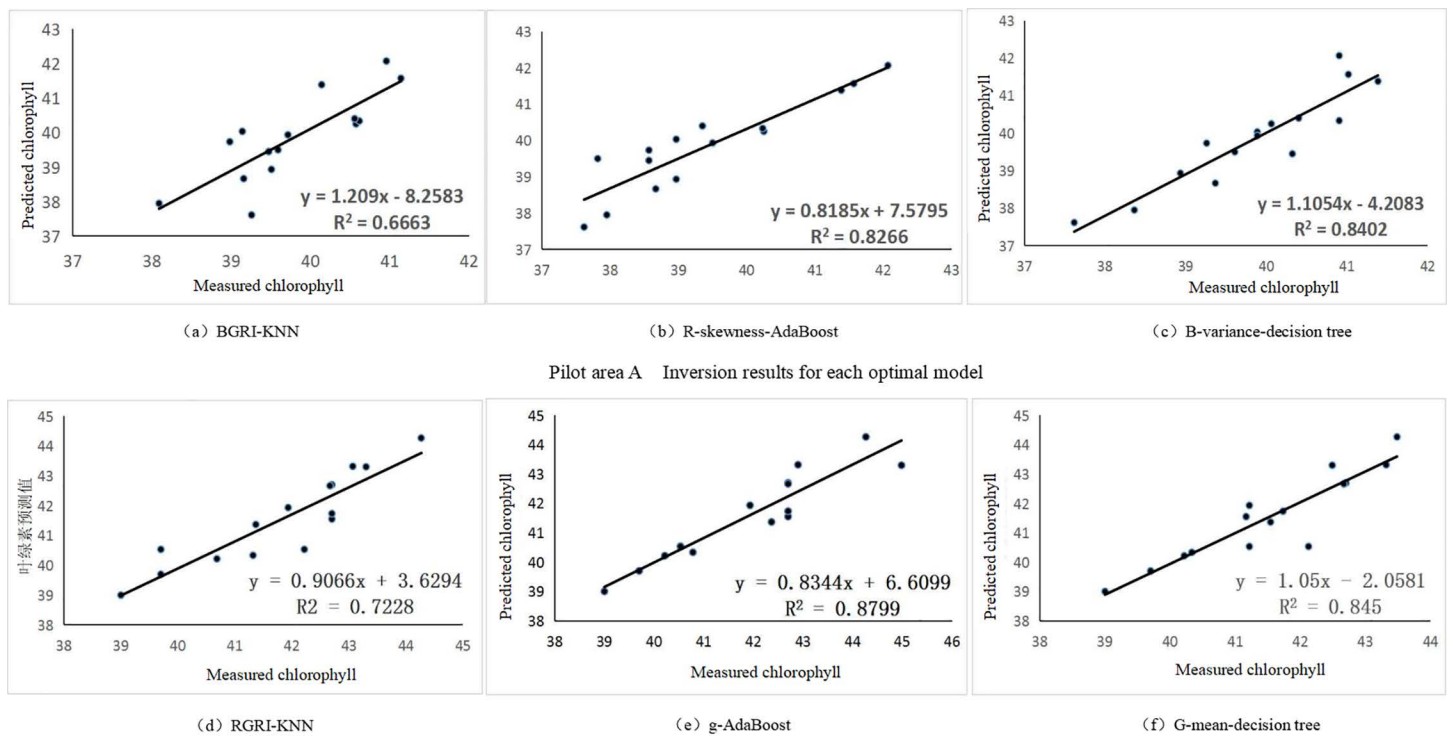

（a）BGRI-KNN　　　　　　　　　（b）R-skewness-AdaBoost　　　　　　　（c）B-variance-decision tree

Pilot area A　　Inversion results for each optimal model

（d）RGRI-KNN　　　　　　　　　（e）g-AdaBoost　　　　　　　　　（f）G-mean-decision tree

Pilot area B Inversion results for each optimal model

**Fig 4. Inversion results of optimal model.**

y = 1.1054x-4.2083, and the fitting accuracy $R^2$ is 0.8402. The best inversion in test area B was the g-AdaBoost model constructed by using the vegetation index g as the independent variable and AdaBoost as the inversion model, with a fitting equation of y = 0.8344x + 6.6099 for the predicted and measured results, and a fitting accuracy $R^2$ of 0.8799. Because there are subtle differences in water content and other organic matter content in soil in different plots, and the data sets collected in different fields have different characteristics, these differences will lead to different performance of models in different data sets, which will affect the selection and performance of models, so the optimal models selected in A and B regions are different. Among the A and B optimal models, the poorer inversion effect is the inversion model constructed with vegetation index and KNN, indicating that the inversion effect of KNN as an inversion model in this visible light rice chlorophyll content inversion has a larger gap than that of the DT and AdaBoost models. This is because the effectiveness of the KNN model depends heavily on the number of training samples. Due to the limited number of rice samples, the model may not be able to fully learn the intrinsic patterns and features in the data, resulting in poor performance of the model in predicting the samples. Finally, combining the accuracy of modeling each model with different variables in subsection 2.2, the best inversion model for test area A was selected as a B-variance-decision tree, with a regression equation of y = 0.002x + 39.466, and the inversion model for test area B was g-AdaBoost, with a regression equation of y = -25.165x + 51.922.

## 3.5. Chlorophyll filling and analysis

The regression equations constructed using the best inversion models B-variance-decision tree and g-AdaBoost in experimental areas A and B were used to fill in the chlorophyll maps of rice at the tasseling stage, and the results of filling in the maps are shown in Fig 5. Fig. 5a

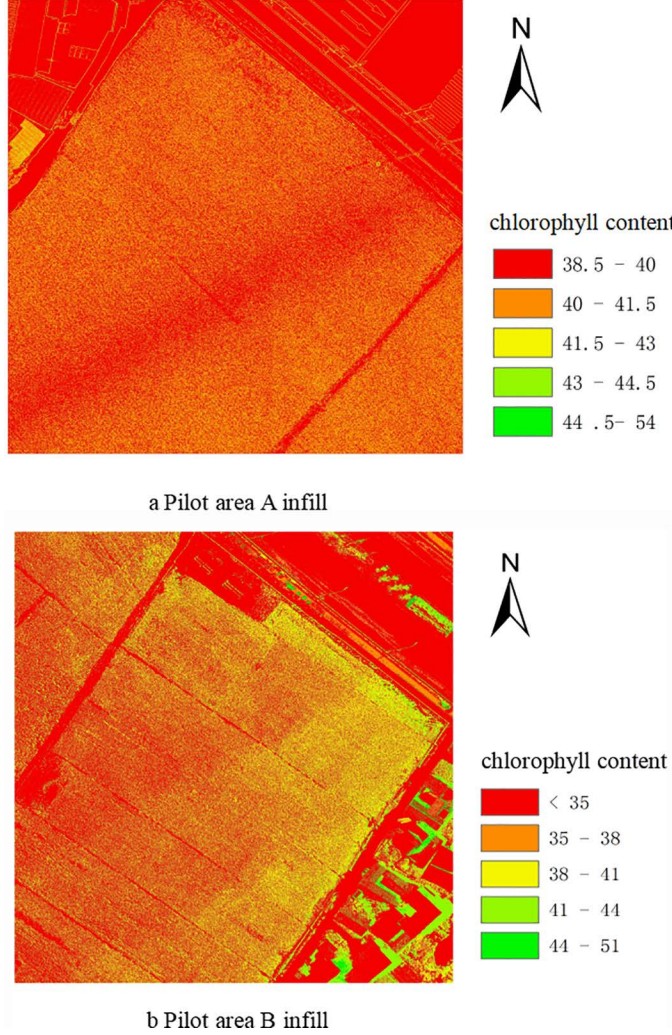

a Pilot area A infill

b Pilot area B infill

**Fig 5. Mapping Of Test Area.**

shows that the chlorophyll values of rice at the tassel stage in experimental area A ranged from 38.5 to 43, with most of them distributed between 40 and 43. In conjunction with Fig 2, it can be seen that the chlorophyll values for the weed areas in the non-rice portion of Area A range from 38.5-40, indicating that within the rice area, a large number of weeds are likely to be present, requiring a manual site visit and targeted weed control. As depicted in Fig 5b, the chlorophyll values for rice during the tapping stage vary within a range of 35-41 within the area. Notably, a significant portion of this area exhibits chlorophyll values predominantly between 35 and 38. A comparative analysis with Fig 2 reveals that there are sections within the area where the chlorophyll levels dip below 35, resembling the chlorophyll concentrations observed in the surrounding weeds. Consequently, it is imperative to conduct a similar field inspection, as done in area A, to assess these areas further. Combining the results of chlorophyll mapping and measured values in the two regions, the predicted chlorophyll results in the test region have a good overlap with the measured results, and the range of values is within the measured data, indicating that the predicted results are reliable. Comparing the two graphs, a and b, the rice is in the same fertility cycle, but there is a large difference in the range

of chlorophyll values, indicating that there is a large difference in growth between the two and that the rice grown manually in region A grows significantly better than the rice grown mechanically in region B. Artificial planting is able to adjust the depth and density of planting according to the specific situation and has flexibility under small fields, which is conducive to controlling the growing environment of rice, thus improving the quality of rice to a certain extent. To make the growth of rice in the two regions converge, the nitrogen, phosphorus and potassium contents of rice in regions A and B were compared and analyzed for scientific and precise fertilization, thus promoting the growth of rice in region B without affecting the development of rice in that region in the subsequent growth cycle.

## 4. Discussions

(1) Extraction of spectral feature bands based on vegetation indices and texture features. Vegetation index and texture feature parameters are important features of visible light images and are commonly used to invert the elemental content of target features [41,42]. In the KNN inversion model, the vegetation index was used as the independent variable in the modeling of the two regions, and the results were better than those of the texture features. In the DT inversion model, the best inversion results are obtained by using texture features as independent variables in the modeling of the two regions, and in the AdaBoost inversion model, the texture features are also used as independent variables in the inversion modeling of the two regions to obtain better results. A comprehensive analysis shows that texture features are more effective for the inversion of rice chlorophyll content. Zhu et al. collected UAV multispectral remote sensing images and ground truth of chlorophyll content using rice in a large field as the research object, and their extracted B_M texture feature had the highest correlation coefficient with rice chlorophyll content, which was 0.73 [43]. In this paper, the correlation coefficient of the texture parameter B-skewness with the rice chlorophyll content was the highest, which was -0.78. To summarize, the correlation coefficients between the texture features on the rice chlorophyll content inversion were higher.

(2) Analysis and comparison of the accuracy of three inversion models. We constructed three inversion models, namely KNN, DT, and AdaBoost. The vegetation index extracted from the two areas with texture features was inputted into the same model, and the accuracy of the obtained models varied greatly, which indicated the different generalization abilities and adaptability of the models [44]. Different models also showed different accuracies among themselves, among which the best inversion was the g-AdaBoost model constructed with the vegetation index g as the independent variable AdaBoost as the inversion model in the experimental area B, with a fitting accuracy of $R^2$ of 0.880. Bai et al. fused three types of feature participants in the construction of the Gaussian regression model to obtain the optimal alfalfa yield estimation accuracy with $R^2 = 0.83$ for the training set, $R^2 = 0.75$ for the validation set, and RPD = 1.98 [45]. The AdaBoost model allows the use of different types of weak learners, such as decision trees, neural networks, etc., which provides the model with greater flexibility and adaptability, which significantly improves the overall model prediction performance [46]. The inversion effect of KNN as an inversion model has a large gap compared to the decision tree and AdaBoost models, indicating that the model is affected by the uneven distribution of training samples, which leads to biased prediction results [47].

(3) Analysis of the applicability of different planting methods. Artificial cultivation of rice is a traditional way of planting. Its advantage is that it can accurately control the position and density of each rice plant, protect rice seedlings, reduce damage, and is suitable for small-scale planting or fields with complex terrain. However, it requires a large amount of labor

cost and time cost and is not suitable for large-scale planting of crops [48]. Mechanical rice transplanting is characterized by high efficiency and speed, which can greatly improve planting efficiency and is suitable for large-scale planting [49]. In addition, it can realize precise row spacing and plant spacing control, which is conducive to the growth and management of rice [50]. Lee et al. used a UAV to acquire multispectral images to develop a new vegetation index, the Square Red and Blue NDVI index, which helps to effectively analyze vegetation in urban areas with various types of land cover, such as long-term steel roofs, waterproof coated roofs, and polyurethane coated areas [51]. Chen et al. used a UAV to capture multispectral images of cotton canopies to extract spectral features and texture features. They utilized Pearson's correlation, principal component analysis, multiple stepwise regression, and ReliefF algorithm for feature selection, and combined with a machine learning algorithm to construct a cotton AGB estimation model [52]. All of the above studies used vegetation indices and texture characterization parameters to invert crop growth, but they were all applied to only one study area. In this paper, two rice fields with different planting methods were selected to participate in the inversion of rice chlorophyll content. The experimental results show that the model inversion accuracy of the mechanically planted area is higher than that of the manually planted area, which indicates that the mechanically planted rice grows uniformly and the distribution of chlorophyll content is more uniform, which is applicable to the Chengdu Plain area, and the experimental results are in agreement with the previous study [53]. Therefore, this study provides a theoretical reference for the subsequent promotion of large-scale mechanical planting of crops and the detection of crop growth.

## 5. Conclusion

The chlorophyll inversion of rice at the tasselling stage was carried out by UAV visible light data using three inversion models, KNN, decision tree and AdaBoost, and the following conclusions can be drawn from the measured data and inversion results:

(1) There was a good correlation between vegetation indices and texture features extracted using UAV visible light imagery and chlorophyll. The highest absolute value of correlation coefficient for vegetation index g is 0.805 in region A; the highest absolute value of correlation coefficient for texture feature B-skewness is 0.83, and the overall correlation of texture features is higher compared to the vegetation index. In Region B as a whole, the correlation coefficients of vegetation index and Region B, as a whole, have lower absolute values of correlation coefficients for vegetation indices and texture features, with the highest absolute value of correlation coefficient for vegetation index g at 0.74, and the highest absolute value of correlation coefficient for texture feature G-mean at 0.55. Taken together, the correlation between vegetation indices and textural characteristics with rice chlorophyll content was better in region A than in region B.

(2) The inversion models constructed based on vegetation indices and texture features with K-neighbourhood (KNN), Decision Tree and AdaBoost, respectively, all achieved better inversion results, and Decision Tree and AdaBoost achieved the best results in the inversion of rice chlorophyll during the tasseling stage in regions A and B, respectively. The best inversion model in region A was identified as a B-variance-decision tree, and the best inversion model in region B was identified as g-AdaBoost.

(3) The best inversion model for each of regions A and B was utilized for map filling. By comparing the results of chlorophyll mapping in the two regions, it can be seen that the growth of rice in Region A is better and more uniform than that in Region B, and the measured

chlorophyll values are less dispersed. By using the chlorophyll inversion mapping, we can recognize the weeds in the rice, and then we can do precise dosing and weeding.

## Author contributions

**Conceptualization:** He Jing.

**Data curation:** Wang Bin.

**Formal analysis:** He Jiachen.

**Methodology:** He Jing, Wang Bin.

**Visualization:** He Jiachen.

**Writing – original draft:** Wang Bin.

**Writing – review & editing:** He Jiachen.

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
