## [Decision Letter · Decision Letter 0]

13 Sep 2024

PONE-D-24-27430Rice Chlorophyll Inversion Based on UAV Visible Light ImageryPLOS ONE

Dear Dr. he,

Thank you for submitting your manuscript to PLOS ONE. After careful consideration, we feel that it has merit but does not fully meet PLOS ONE’s publication criteria as it currently stands. Therefore, we invite you to submit a revised version of the manuscript that addresses the points raised during the review process.

The manuscript must be corrected in all points indicated by the reviewers, such as:

1) The author's title does not effectively summarise the work.

2) The authors do not provide a clear description of the research objectives in the introduction.

3) The marginal contributions, especially the innovative nature of the manuscript, are not well written.

4) The objectives of the study are not clear.

5) The paper lacks discussion, especially with other similar methods. 6) The discussion should indicate the advantages and disadvantages of the method, especially applicability, cost, and detection accuracy.

We look forward to receiving your revised manuscript.

Kind regards,

Claudionor Ribeiro da Silva

Academic Editor

PLOS ONE

Journal Requirements: When submitting your revision, we need you to address these additional requirements. 1. Please ensure that your manuscript meets PLOS ONE's style requirements, including those for file naming. The PLOS ONE style templates can be found at https://journals.plos.org/plosone/s/file?id=wjVg/PLOSOne_formatting_sample_main_body.pdf and https://journals.plos.org/plosone/s/file?id=ba62/PLOSOne_formatting_sample_title_authors_affiliations.pdf 2. Please update your submission to use the PLOS LaTeX template. The template and more information on our requirements for LaTeX submissions can be found at http://journals.plos.org/plosone/s/latex. 3. In your Methods section, please provide additional information regarding the permits you obtained for the work. Please ensure you have included the full name of the authority that approved the field site access and, if no permits were required, a brief statement explaining why. 4. Please note that PLOS ONE has specific guidelines on code sharing for submissions in which author-generated code underpins the findings in the manuscript. In these cases, we expect all author-generated code to be made available without restrictions upon publication of the work. Please review our guidelines at https://journals.plos.org/plosone/s/materials-and-software-sharing#loc-sharing-code and ensure that your code is shared in a way that follows best practice and facilitates reproducibility and reuse. 5. Thank you for stating the following financial disclosure: "Chengdu City Technology Innovation R&D Project (2022-YF05-01090-SN)" Please state what role the funders took in the study.  If the funders had no role, please state: ""The funders had no role in study design, data collection and analysis, decision to publish, or preparation of the manuscript."" If this statement is not correct you must amend it as needed. Please include this amended Role of Funder statement in your cover letter; we will change the online submission form on your behalf. 6. We note that your Data Availability Statement is currently as follows: Indicate whether you have had any of the following previous interactions about this manuscript Please confirm at this time whether or not your submission contains all raw data required to replicate the results of your study. Authors must share the “minimal data set” for their submission. PLOS defines the minimal data set to consist of the data required to replicate all study findings reported in the article, as well as related metadata and methods (https://journals.plos.org/plosone/s/data-availability#loc-minimal-data-set-definition). For example, authors should submit the following data: - The values behind the means, standard deviations and other measures reported;- The values used to build graphs;- The points extracted from images for analysis. Authors do not need to submit their entire data set if only a portion of the data was used in the reported study. If your submission does not contain these data, please either upload them as Supporting Information files or deposit them to a stable, public repository and provide us with the relevant URLs, DOIs, or accession numbers. For a list of recommended repositories, please see https://journals.plos.org/plosone/s/recommended-repositories. If there are ethical or legal restrictions on sharing a de-identified data set, please explain them in detail (e.g., data contain potentially sensitive information, data are owned by a third-party organization, etc.) and who has imposed them (e.g., an ethics committee). Please also provide contact information for a data access committee, ethics committee, or other institutional body to which data requests may be sent. If data are owned by a third party, please indicate how others may request data access. 7. Please amend the manuscript submission data (via Edit Submission) to include author Dr. Wang Bin and Dr. He Jiachen. 8. We note that Figures 1 and 2 in your submission contain [map/satellite] images which may be copyrighted. All PLOS content is published under the Creative Commons Attribution License (CC BY 4.0), which means that the manuscript, images, and Supporting Information files will be freely available online, and any third party is permitted to access, download, copy, distribute, and use these materials in any way, even commercially, with proper attribution. For these reasons, we cannot publish previously copyrighted maps or satellite images created using proprietary data, such as Google software (Google Maps, Street View, and Earth). For more information, see our copyright guidelines: http://journals.plos.org/plosone/s/licenses-and-copyright">http://journals.plos.org/plosone/s/licenses-and-copyright. We require you to either (1) present written permission from the copyright holder to publish these figures specifically under the CC BY 4.0 license, or (2) remove the figures from your submission: A. You may seek permission from the original copyright holder of Figures 1 and 2 to publish the content specifically under the CC BY 4.0 license.   We recommend that you contact the original copyright holder with the Content Permission Form (http://journals.plos.org/plosone/s/file?id=7c09/content-permission-form.pdf) and the following text:“I request permission for the open-access journal PLOS ONE to publish XXX under the Creative Commons Attribution License (CCAL) CC BY 4.0 (http://creativecommons.org/licenses/by/4.0/). Please be aware that this license allows unrestricted use and distribution, even commercially, by third parties. Please reply and provide explicit written permission to publish XXX under a CC BY license and complete the attached form.” Please upload the completed Content Permission Form or other proof of granted permissions as an ""Other"" file with your submission. In the figure caption of the copyrighted figure, please include the following text: “Reprinted from [ref] under a CC BY license, with permission from [name of publisher], original copyright [original copyright year].” B. If you are unable to obtain permission from the original copyright holder to publish these figures under the CC BY 4.0 license or if the copyright holder’s requirements are incompatible with the CC BY 4.0 license, please either i) remove the figure or ii) supply a replacement figure that complies with the CC BY 4.0 license. Please check copyright information on all replacement figures and update the figure caption with source information. If applicable, please specify in the figure caption text when a figure is similar but not identical to the original image and is therefore for illustrative purposes only.The following resources for replacing copyrighted map figures may be helpful: USGS National Map Viewer (public domain): http://viewer.nationalmap.gov/viewer/The Gateway to Astronaut Photography of Earth (public domain): http://eol.jsc.nasa.gov/sseop/clickmap/Maps at the CIA (public domain): https://www.cia.gov/library/publications/the-world-factbook/index.html and https://www.cia.gov/library/publications/cia-maps-publications/index.htmlNASA Earth Observatory (public domain): http://earthobservatory.nasa.gov/Landsat: http://landsat.visibleearth.nasa.gov/USGS EROS (Earth Resources Observatory and Science (EROS) Center) (public domain): http://eros.usgs.gov/#Natural Earth (public domain): http://www.naturalearthdata.com/

Reviewers' comments:

Reviewer's Responses to Questions

**Comments to the Author**

1. Is the manuscript technically sound, and do the data support the conclusions?

Reviewer #1: Yes

Reviewer #2: Yes

2. Has the statistical analysis been performed appropriately and rigorously? 

Reviewer #1: Yes

Reviewer #2: Yes

3. Have the authors made all data underlying the findings in their manuscript fully available?

Reviewer #1: Yes

Reviewer #2: Yes

4. Is the manuscript presented in an intelligible fashion and written in standard English?

Reviewer #1: Yes

Reviewer #2: Yes

5. Review Comments to the Author

Reviewer #1: The author's title does not effectively summarise the work, and "Rice growth monitoring model and application based on UAV visible chlorophyll inversion" seems to be more in line with the author's research.

The authors do not provide a clear description of the research objectives in the introduction, and the marginal contributions, especially the innovative nature of the manuscript, are not well written. In addition, the objectives of the study are not clear.

In addition, the authors state that the data were collected on August day of 2021, but they did not collect and process the data for the similar time period of 2022 and 2023, so please explain the rationale. Is it related to weather conditions? The robustness of the findings leaves something to be desired as the manuscript did not repeat the experiment for the final results.

The presentation of Figure 2 confuses the reader, with no indication of where the arrows point to in terms of bearing and location in regions A and B, respectively.

The paper lacks discussion, especially with other similar methods. The discussion should indicate the advantages and disadvantages of the method, especially applicability, cost, and detection accuracy. The current manuscript is more like a summary and generalisation of an experimental procedure.

The authors' references are not standardised. References in Chinese should be labelled “(In Chinese)”.

Reviewer #2: Try to provide citations in the introduction, background, and methodology. Are there any gaps with previous research? Provide at least the last 5 years for reference. And is there any continuation of this study?

6. PLOS authors have the option to publish the peer review history of their article (what does this mean? ). If published, this will include your full peer review and any attached files.

**Do you want your identity to be public for this peer review?** For information about this choice, including consent withdrawal, please see our Privacy Policy .

Reviewer #1: No

Reviewer #2: No

---

## [Author Response · Author response to Decision Letter 0]

4 Nov 2024

The manuscript must be corrected in all points indicated by the reviewers, such as:

1) The author's title does not effectively summarise the work.

Response: Thank you very much for the suggestion. We have changed the title to ‘Spectral information and texture features synergize for rice chlorophyll inversion’ to better summarise the work of this paper.

2) The authors do not provide a clear description of the research objectives in the introduction.

Response:Thank you very much for the suggestion. We have revised the introduction to add a clear description of the research objectives, as described below: Therefore, in this paper, we extracted the spectral information and texture features of UAV visible images, constructed 14 spectral indices and 15 texture features, and selected the feature variables that were highly correlated with the measured rice chlorophyll content as the input variables of the model using correlation analysis. We compared the inversion results of these two parameters in three different models, namely, K-neighborhood (KNN), Decision Tree (DT), and AdaBoost, and evaluated the performance differences among these three inversion models in predicting the chlorophyll content of rice, and selected the best inversion model. In addition, to explore the applicability of the models in different planting types of farmland, we selected two representative planting types of farmland (manual planting and mechanical planting). We tested the inversion results of the three models in these two types of farmland, to deeply explore the applicability conditions of the models as well as the influence of the planting types of cropland on the inversion results. It provides the theoretical basis and practical guidance for the inversion of chlorophyll content of rice at the tasseling stage by UAV visible light images.

3) The marginal contributions, especially the innovative nature of the manuscript, are not well written.

Response: Thank you very much for the suggestion. For the description of the paper's innovative nature, we have revised and summarized it in the introduction section as follows: We compared the inversion results of these two parameters in three different models, namely, K-neighborhood (KNN), Decision Tree (DT), and AdaBoost, and evaluated the performance differences among these three inversion models in predicting the chlorophyll content of rice, and selected the best inversion model. In addition, to explore the applicability of the models in different planting types of farmland, we selected two representative planting types of farmland (manual planting and mechanical planting). We tested the inversion results of the three models in these two types of farmland, to deeply explore the applicability conditions of the models as well as the influence of the planting types of cropland on the inversion results. It provides the theoretical basis and practical guidance for the inversion of chlorophyll content of rice at the tasseling stage by UAV visible light images.

4) The objectives of the study are not clear.

Response: Thank you very much for the suggestion. The goal of our research is to invert the chlorophyll content of rice leaves at the tasseling stage using vegetation indices and texture feature parameters combined with various machine learning algorithms, to select the optimal inversion model, and to explore the planting method suitable for Chengdu's farmland, so as to provide a theoretical basis for the inversion of chlorophyll content of rice at the tasseling stage using UAV visible light imagery. This content we have modified in the introduction part.

5) The paper lacks discussion, especially with other similar methods.

Response: Thank you very much for the suggestion. We have added a discussion section to explore the three parts of characterization parameters, inversion models and planting methods. And we compared with other similar research methods to highlight the innovation of this paper.

6) The discussion should indicate the advantages and disadvantages of the method, especially applicability, cost, and detection accuracy.

Response: Thank you very much for the suggestion. In the first subsection of our discussion, we explored the effect of inversion of the vegetation index with the texture feature parameter and verified the generalization of these two parameters across different cropping patterns. We explore the generalization ability and applicability of the model in the second subsection of the discussion. the AdaBoost model allows the use of different types of weak learners, which provides the model with greater flexibility and adaptability, and significantly improves the prediction performance of the overall model. the inversion effect of KNN as an inversion model has a larger gap than that of the decision tree and the AdaBoost model, which suggests that the model is affected by the non-uniformity of the distribution of the training samples, resulting in biased prediction results. uniformity of the training samples, resulting in biased prediction results. In the third subsection of our discussion, we explore the advantages and disadvantages of manual versus mechanical planting, and analyze planting methods suitable for the farmland of Chengdu.

Response: Thank you very much for the suggestion. We have revised the manuscript meets according to the PLOS ONE's style requirements.

Response: Thank you very much for the suggestion. We have revised the manuscript meets according to the PLOS ONE's style requirements.

Response: Thank you very much for the suggestion. We were not required to provide permits for our work because the study area we visited was part of our contracted test plots.

4. Please note that PLOS ONE has specific guidelines on code sharing for submissions in which author-generated code underpins the findings in the manuscript. In these cases, we expect all author-generated code to be made available without restrictions upon publication of the work. Please review our guidelines at https://journals.plos.org/plosone/s/materials-and-software-sharing#loc-sharing-code and ensure that your code is shared in a way that follows best practice and facilitates reproducibility and reuse.

Response: Thank you very much for the suggestion. We code is shared in a way that follows best practice and facilitates reproducibility and reuse.

5. Thank you for stating the following financial disclosure: "Chengdu City Technology Innovation R&D Project (2022-YF05-01090-SN)" Please state what role the funders took in the study. If the funders had no role, please state: "The funders had no role in study design, data collection and analysis, decision to publish, or preparation of the manuscript." "If this statement is not correct you must amend it as needed. Please include this amended Role of Funder statement in your cover letter; we will change the online submission formon your behalf.

Response: Thank you very much for the suggestion. The funders had no role in study design, data collection and analysis, decision to publish, or preparation of the manuscript. And we have deleted that sentence.

6. We note that your Data Availability Statement is currently as follows: Indicate whether you have had any of the following previous interactions about this manuscript. Please confirm at this time whether or not your submission contains all raw data required to replicate the results of your study. Authors must share the “minimal data set” for their submission. PLOS defines the minimal data set to consist of the data required to replicate all study findings reported in the article, as well as related metadata and methods (https://journals.plos.org/plosone/s/data-availability#loc-minimal-data-set-definition).

Response: Thank you very much for the suggestion. Our submission contains all the raw data needed to replicate the results of the study. Our research data is original and we have removed this statement.

7. Please amend the manuscript submission data (via Edit Submission) to include author Dr. Wang Bin and Dr. He Jiachen.

Response: Thank you very much for the suggestion. We will amend the manuscript submission data (via Edit Submission) to include author Dr. Wang Bin and Dr. He Jiachen.

8. We note that Figures 1 and 2 in your submission contain [map/satellite] images which may be copyrighted. All PLOS content is published under the Creative Commons Attribution License (CC BY 4.0), which means that the manuscript, images, and Supporting Information files will be freely available online, and any third party is permitted to access, download, copy, distribute, and use these materials in any way, even commercially, with proper attribution. For these reasons, we cannot publish previously copyrighted maps or satellite images created using proprietary data, such as Google software (Google Maps, Street View, and Earth). For more information, see our copyright guidelines: http://journals.plos.org/plosone/s/licenses-and-copyright.

Response: Thank you very much for the suggestion. The Figures 1 and 2 of the paper are visible light images taken by us using a drone and do not contain copyrighted images.

---

## [Decision Letter · Decision Letter 1]

26 Nov 2024

PONE-D-24-27430R1Spectral information and texture features synergize for rice chlorophyll inversionPLOS ONE

Dear Dr. he,

Thank you for submitting your manuscript to PLOS ONE. After careful consideration, we feel that it has merit but does not fully meet PLOS ONE’s publication criteria as it currently stands. Therefore, we invite you to submit a revised version of the manuscript that addresses the points raised during the review process.

The manuscript must be corrected in all points indicated by the reviewers, such as:

1. The authors have made the necessary changes to the title, but it still does not give a complete picture of their work and innovations. It is recommended that the authors further revise the title to ensure that it completely summarizes the manuscript.

2. The author's introduction has been improved by adding citations of references and a description of the innovation, but the revision is still not in place. The introduction lacks a description of the clear relevance of inverting chlorophyll content in rice at the tassel stage. The authors should have written a clear comparison between the shortcomings of the current research methodology and the innovations of the manuscript.

3. The authors have added a discussion section. Still, the discussion section lacks reference citations, and the authors should cite the results of other research methods to argue the reliability of the experimental results of the paper. In addition, it is recommended that the authors add a subheading in front of each paragraph of the discussion and make clear the relevance of each part of the discussion to the topic.

4. The conclusion section should summarize the discussion surrounding the article by appropriately describing its conclusion rather than repeating the statement of experimental results.

5. The authors should increase the line numbering of the manuscript to facilitate reviewers' access.

We look forward to receiving your revised manuscript.

Kind regards,

Claudionor Ribeiro da Silva

Academic Editor

PLOS ONE

Journal Requirements:

Reviewers' comments:

Reviewer's Responses to Questions

**Comments to the Author**

1. If the authors have adequately addressed your comments raised in a previous round of review and you feel that this manuscript is now acceptable for publication, you may indicate that here to bypass the “Comments to the Author” section, enter your conflict of interest statement in the “Confidential to Editor” section, and submit your "Accept" recommendation.

Reviewer #1: All comments have been addressed

Reviewer #2: All comments have been addressed

2. Is the manuscript technically sound, and do the data support the conclusions?

Reviewer #1: Yes

Reviewer #2: Yes

3. Has the statistical analysis been performed appropriately and rigorously? 

Reviewer #1: Yes

Reviewer #2: Yes

4. Have the authors made all data underlying the findings in their manuscript fully available?

Reviewer #1: Yes

Reviewer #2: Yes

5. Is the manuscript presented in an intelligible fashion and written in standard English?

Reviewer #1: Yes

Reviewer #2: Yes

6. Review Comments to the Author

Reviewer #1: 1. The authors have made the necessary changes to the title, but it still does not give a complete picture of their work and innovations. It is recommended that the authors further revise the title to ensure that it completely summarizes the manuscript.

2. The author's introduction has been improved by adding citations of references and a description of the innovation, but the revision is still not in place. The introduction lacks a description of the clear relevance of inverting chlorophyll content in rice at the tassel stage. The authors should have written a clear comparison between the shortcomings of the current research methodology and the innovations of the manuscript.

3. The authors have added a discussion section. Still, the discussion section lacks reference citations, and the authors should cite the results of other research methods to argue the reliability of the experimental results of the paper. In addition, it is recommended that the authors add a subheading in front of each paragraph of the discussion and make clear the relevance of each part of the discussion to the topic.

4. The conclusion section should summarize the discussion surrounding the article by appropriately describing its conclusion rather than repeating the statement of experimental results.

5. The authors should increase the line numbering of the manuscript to facilitate reviewers' access.

Reviewer #2: The authors declare that they have no known competing financial interests or personal

relationships that could have appeared to influence the work reported in this paper. The chlorophyll inversion of rice at the tasselling stage was carried out by UAV visible light data using three inversion models, KNN, decision tree and AdaBoost, and the following conclusions can be drawn from the measured data and inversion results

7. PLOS authors have the option to publish the peer review history of their article (what does this mean? ). If published, this will include your full peer review and any attached files.

**Do you want your identity to be public for this peer review?** For information about this choice, including consent withdrawal, please see our Privacy Policy .

Reviewer #1: No

Reviewer #2: No

---

## [Author Response · Author response to Decision Letter 1]

30 Dec 2024

Dear Editor:

Manuscript ID PONE-D-24-27430 entitled "Chlorophyll inversion in rice based on visible light images of different planting methods" which has been revised.

We are truly grateful to yours critical comments and thoughtful suggestions. Based on these comments and suggestions, we have made careful modifications on the manuscript, including title, introduction, and discussion. All changes made to the text are in red color.

We hope the new manuscript will meet your magazine’s standard. Please do forward our heartfelt thanks to these experts. We look forward to hearing from you soon for a favorable decision.

We tried our best to improve the manuscript and made a significant change in the manuscript. Below you will find our point-by-point responses to the reviewers’ and editor’s comments/ questions.

Comments from the editors and reviewers:

The manuscript must be corrected in all points indicated by the reviewers, such as:

1)The authors have made the necessary changes to the title, but it still does not give a complete picture of their work and innovations. It is recommended that the authors further revise the title to ensure that it completely summarizes the paper's work.

Response: Thank you very much for the suggestion. We have changed the title to ‘Chlorophyll inversion in rice based on visible light images of different planting methods’ to better summarise the work of this paper.

2)The author's introduction has been improved by adding citations of references and a description of the innovation, but the revision is still not in place. The introduction lacks a description of the clear relevance of inverting chlorophyll content in rice at the tassel stage. The authors should have written a clear comparison between the shortcomings of the current research methodology and the innovations of the paper.

Response:Thank you very much for the suggestion. Regarding the lack of a clear description of the significance of the inversion of chlorophyll content at the rice tasseling stage in the introduction, we have made the following changes: By determining the chlorophyll content of rice during the spiking period, it is possible to determine whether rice growth is healthy or not, to intervene earlier in the growth of rice during the critical period, and to predict its future yield, thus providing a basis for subsequent crop management. In the introduction, we summarize the shortcomings of existing research methods and present the innovations in our experiments, modified as follows: “In recent years, the mechanization of rice production has shown a rapid development trend, the proportion of manual planting has decreased, and mechanized rice transplanting is gradually replacing traditional manual transplanting. As far as rice planting is concerned, replacing the traditional manual planting with the simple and efficient planting method represented by mechanical transplanting is an important way to improve the efficiency of rice production and reduce the cost of human labor [16]. Numerous scholars have evaluated rice yield and economic efficiency [17], microbial populations [18], pest and weed pest occurrence characteristics [19-20], nitrogen utilization efficiency [21], and greenhouse gas emissions [22] under different planting methods, and concluded that different planting methods have different rice yield levels, nutrient uptake and utilization, and economic benefits have certain differences. However, relatively few studies have been conducted to evaluate the chlorophyll content of rice paddies under different planting methods in the plains.” “Therefore, this paper proposes a chlorophyll inversion model that synergizes visible spectral information and texture features, which can invert the chlorophyll content of rice more accurately and efficiently.” “We tested the inversion effects of the three models in these two types of farmland and deeply explored the applicable conditions of the models and the effects of different planting methods on the inversion effects of rice chlorophyll content in the plain area. This will provide a low-cost and high-precision monitoring method and reference basis for the development process of mechanized agriculture.”

3)The authors have added a discussion section. Still, the discussion section lacks reference citations, and the authors should cite the results of other research methods to argue the reliability of the experimental results of the paper. In addition, it is recommended that the authors add a subheading in front of each paragraph of the discussion and make clear the relevance of each part of the discussion to the topic.

Response: Thank you very much for the suggestion. We have cited other research methods and compared them with the experimental results in this paper, modified as follows: “Zhu et al. collected UAV multispectral remote sensing images and ground truth of chlorophyll content using rice in a large field as the research object, and their extracted B_M texture feature had the highest correlation coefficient with rice chlorophyll content, which was 0.73 [43]. In this paper, the correlation coefficient of the texture parameter B-skewness with the rice chlorophyll content was the highest, which was -0.78. To summarize, the correlation coefficients between the texture features on the rice chlorophyll content inversion were higher.” “Bai et al. fused three types of feature participants in the construction of the Gaussian regression model to obtain the optimal alfalfa yield estimation accuracy with R2 = 0.83 for the training set, R2 = 0.75 for the validation set, and RPD = 1.98.” “All of the above studies used vegetation indices and texture characterization parameters to invert crop growth, but they were all applied to only one study area. In this paper, two rice fields with different planting methods were selected to participate in the inversion of rice chlorophyll content. The experimental results show that the model inversion accuracy of the mechanically planted area is higher than that of the manually planted area, which indicates that the mechanically planted rice grows uniformly and the distribution of chlorophyll content is more uniform, which is applicable to the Chengdu Plain area, and the experimental results are in agreement with the previous study”. In addition, we have added subheadings to each paragraph of the discussion.

4)The conclusion section should summarize the discussion surrounding the article by appropriately describing its conclusion rather than repeating the statement of experimental results.

Response: Thank you very much for the suggestion. We removed the description of the reproducibility of the experimental results and condensed the original five conclusions into three.

5)The authors should increase the line numbering of the paper to facilitate reviewers' access.

Response: Thank you very much for the suggestion. We have added line numbers to the paper.

---

## [Decision Letter · Decision Letter 2]

6 Feb 2025

Chlorophyll inversion in rice based on visible light images of different planting methods

PONE-D-24-27430R2

Dear Dr. he,

We’re pleased to inform you that your manuscript has been judged scientifically suitable for publication and will be formally accepted for publication once it meets all outstanding technical requirements.

Kind regards,

Claudionor Ribeiro da Silva

Academic Editor

PLOS ONE

Additional Editor Comments (optional):

Reviewers' comments:

Reviewer's Responses to Questions

**Comments to the Author**

1. If the authors have adequately addressed your comments raised in a previous round of review and you feel that this manuscript is now acceptable for publication, you may indicate that here to bypass the “Comments to the Author” section, enter your conflict of interest statement in the “Confidential to Editor” section, and submit your "Accept" recommendation.

Reviewer #1: All comments have been addressed

Reviewer #2: (No Response)

2. Is the manuscript technically sound, and do the data support the conclusions?

Reviewer #1: Yes

Reviewer #2: Yes

3. Has the statistical analysis been performed appropriately and rigorously? 

Reviewer #1: Yes

Reviewer #2: Yes

4. Have the authors made all data underlying the findings in their manuscript fully available?

Reviewer #1: Yes

Reviewer #2: Yes

5. Is the manuscript presented in an intelligible fashion and written in standard English?

Reviewer #1: Yes

Reviewer #2: Yes

6. Review Comments to the Author

Reviewer #1: (No Response)

Reviewer #2: I think that the article was in good form and have concenr to her science discipline, nothing else dual publication, has a research ethics. Overall, I think it must be proceed

7. PLOS authors have the option to publish the peer review history of their article (what does this mean? ). If published, this will include your full peer review and any attached files.

**Do you want your identity to be public for this peer review?** For information about this choice, including consent withdrawal, please see our Privacy Policy .

Reviewer #1: No

Reviewer #2: **Yes: ** Ghefra Rizkan Gaffara

---

## [Editor Report · Acceptance letter]

PONE-D-24-27430R2

PLOS ONE

Dear Dr. Jing,

I'm pleased to inform you that your manuscript has been deemed suitable for publication in PLOS ONE. Congratulations! Your manuscript is now being handed over to our production team.

Kind regards,

on behalf of

Dr. Claudionor Ribeiro da Silva

Academic Editor

PLOS ONE